# Comparison of Three Different Techniques for the Treatment of Cartilage Lesions—Matrix-Induced Autologous Chondrocyte Implantation (MACI) Versus Autologous Matrix-Induced Chondrogenesis (AMIC) and Arthroscopic Minced Cartilage—A 2-Year Follow-Up on Patient-Reported Pain and Functional Outcomes

**DOI:** 10.3390/jcm14072194

**Published:** 2025-03-23

**Authors:** Stefan Schneider, Dagmar Linnhoff, Ansgar Ilg, Gian M. Salzmann, Robert Ossendorff, Johannes Holz

**Affiliations:** 1OrthoCentrum Hamburg, 20149 Hamburg, Germany; dr.ilg@oc-h.de (A.I.); dr.holz@oc-h.de (J.H.); 2Gelenkzentrum Rhein-Main, 65239 Hochheim am Main, Germany; salzmann.info@kws.ch; 3Schulthess Klinik, 8008 Zurich, Switzerland; 4Department of Orthopaedics and Trauma Surgery, University Hospital Bonn, 53127 Bonn, Germany; robert.ossendorff@ukbonn.de; 5MSH Medical School Hamburg, University of Applied Sciences and Medical University, 20457 Hamburg, Germany

**Keywords:** cartilage lesion, regenerative therapy, orthobiologics

## Abstract

**Background/Objectives**: The treatment of cartilage damage is an ongoing challenge. Several techniques have been developed to address this problem. Matrix-Induced Autologous Chondrocyte Implantation (MACI) is often referred to as the “gold standard” for cartilage treatment. Numerous long-term outcome studies also have reported favorable results with Autologous Matrix-Induced Chondrogenesis (AMIC). Minced Cartilage Implantation (MCI) is a recently developed arthroscopic method. This technique has demonstrated promising outcomes, with the prospect of longer-term results still under investigation. This study aims to directly compare the patient-reported outcomes of these three techniques over a 2-year follow-up period. **Methods**: A total of *N* = 48 patients were included in the retrospective matched pair analysis (*n* = 16 MACI, *n* = 16 AMIC, *n* = 16 MCI). VAS, KOOS-Pain, and KOOS-Symptoms scores served as primary outcomes; the KOOS-ADL and -QOL and the Tegner Activity Scale (TAS) served as secondary outcomes. **Results**: All three groups did not differ from each other in the primary or secondary outcomes. Pain and function significantly improved from pre-surgery to two years after (VAS: *p* < 0.000; ES: η^2^ = 0.27; KOOS-Pain: *p* < 0.000; ES: η^2^ = 0.30; KOOS-Symptoms: *p* = 0.000; ES: η^2^ = 0.26; KOOS-ADL: *p* > 0.000; ES: η^2^ = 0.20; KOOS-QOL: *p* > 0.000; ES: η^2^ = 0.30). There was no significant effect of time on the activity level. **Conclusions**: All three procedures show good patient-reported outcomes, low complication rates, and long graft longevity in the 2-year follow-up. Therefore, all three methods seem to be equally recommendable for the treatment of cartilage lesions.

## 1. Introduction

Articular cartilage has a crucial role in joints by absorbing pressure, rendering it elastic to compression and flexion and enabling pain-free movement. During osteoarthritis progression, mainly in the elderly population, articular cartilage is significantly affected by degeneration. Studies have shown that osteoarthritic changes also exist in the younger population, caused by various factors like body constitution, genetics, personal lifestyle, or working environment [1]. This makes the degeneration of articular cartilage a worldwide socioeconomic problem [2].

Articular cartilage is composed of chondrocytes and an extracellular matrix. Chondrocytes have a special function in the formation, maintenance, and repair of the extracellular matrix [3]. It comprises various connective and supportive elements, such as proteoglycans and collagens, and is characterized by a scarcity of blood vessels. Nutrient diffusion from synovial fluid is the primary means of nourishing cartilage tissue [4]. The architecture can be subdivided into zones with different biomechanical entities. At the superficial zone, collagen fibers with small diameters are oriented parallel to the surface, allowing load-induced stresses to be compensated [5].

In individuals who engage in regular physical activity, the cartilage must endure loads equivalent to 10 to 20 times their body weight. Due to its mechanical properties, normal, physiological cartilage can withstand higher pressures. The surface properties of the superficial zone, in particular, are responsible for the mechanical reactions of the cartilage [6]. With the onset of degeneration, these mechanical properties change fundamentally [6,7]. Degenerated cartilage becomes softer, less resilient, and more permeable due to the loss of extracellular matrix integrity. The cartilage’s ability to withstand compression drops significantly while permeability increases [7,8]. This significantly impairs weight-bearing, stabilizing, and lubricating functions. Progressively reduced cartilage stiffness and a decreased ability to resist deformation finally results in structural failure. A detailed report elaborating the mechanical aspects of changes and repair in articular cartilage was recently published by Krakowski and colleagues [7].

Hyaline articular cartilage exhibits limited inherent self-repair capabilities. Chondrocytes are unable to efficiently multiply and migrate for the generation of high-quality repair tissue, whether the damage is superficial or deep [9,10]. Traumatic or degenerative damage typically results in the development of repair tissue that possesses inferior structural and biomechanical attributes compared to healthy articular cartilage [4]. The current focus in managing symptomatic cartilage defects in the knee joint revolves around the repair of the articular surface, the restoration of joint balance, and, ultimately, the prevention of osteoarthritis progression.

In the late 1990s, the commonly employed therapeutic options included surface debridement procedures, like chondral shaving, abrasion chondroplasty, and subchondral perforation, as well as soft tissue arthroplasty procedures utilizing perichondral or periosteal grafts. However, none of these methods yielded normal, hyaline-like cartilage tissue. More recent techniques based on bone marrow stimulation or cell application have since been introduced. Microfracture (MFX) has been widely used in patients since 1998 and involves creating perforations in the subchondral bone, spaced 3 to 4 mm apart, to release bone marrow components into the defect. To date, MFX stands as a widely adopted bone marrow stimulation strategy, and a range of cell-based restoration techniques utilizing adult or juvenile and autograft or allograft cartilage sources, along with chondrocyte and nonchondrocyte options, are available [11].

Historically, the gold standard for repairing cartilage defects has been Matrix-Induced Autologous Chondrocyte Implantation (MACI). Dr. Lars Peterson pioneered this treatment method in 1985, which initially faced skepticism but later achieved notable success in 1989. In MACI, cartilage cells are harvested in a first surgical procedure, cultivated in the laboratory, and then implanted in a second procedure. In 2000, the United Kingdom’s National Institute for Health and Care Excellence did not recommend routine use of the procedure due to a lack of well-designed studies. In 2005, and later, in 2017, positive clinical and economic evaluations were available [12,13]. MACI’s ability to improve clinical outcomes is now supported by numerous studies, including a minimum 10-year outcome systematic review conducted in 2024 [14]. The rate of surgical technical errors has significantly decreased from the first to the second generations of this technique (from 26% to 0.5%), and the 5-year survival rates currently stand at 78% [15]. It is now evident that factors such as lesion size, prior disease, surgical course, and post-treatment symptoms play significant roles in the survival rates [16].

An alternative approach to treating cartilage lesions is the one-stage matrix-based treatment known as Autologous Matrix-Induced Chondrogenesis (AMIC), developed in 2005 [17,18]. In this technique, a matrix is implanted after microfracturing has been performed. The matrix provides a temporary structure to support cell migration and cartilage formation. This contrasts with MACI, where intact cartilage is arthroscopically removed from an unaffected area of the damaged cartilage in a first surgical procedure and then cultivated on a matrix in vitro in a second step [19].

In AMIC, periosteal hypertrophy may occur between three and seven months after surgery in 10% to 25% of cases, often necessitating revision surgery [20].

Another emerging technique that has garnered significant interest in treating cartilage lesions is Minced Cartilage Implantation (MCI). In 1983, Albrecht, Roessner, and Zimmermann first described the treatment of osteochondral lesions with autologous chondral fragments and fibrin glue [21]. Articles describing similar techniques have proliferated over the last five years. Preclinical in vivo and in vitro data have shown promising results regarding chondrocyte activation through fragmentation [16,22]. Cartilage mincing leads to the outgrowth, proliferation, and differentiation of articular chondrocytes. Since these proliferation and differentiation processes occur intraarticularly, the cells are consistently exposed to a native physical and biochemical environment [23].

In the initial descriptions of the MCI technique, cartilage fragments were manually cut into small pieces and then secured in the defect using a membrane or allogeneic fibrin glue. For this open procedure, a 5-year follow-up could present positive patient-related outcomes and low complication rates [24]. This technique has since evolved towards an arthroscopic, entirely autologous approach, with the first technical notes published by Schneider and Salzmann in 2020 [16].

Although the three procedures (MACI, AMIC, and MCI) differ in technical aspects, time, and cost, as previously described, they have all shown good outcomes in the past. Nevertheless, it is worth investigating whether a direct comparison of the three procedures reveals any significant differences in the results reported by patients. To date, no study has directly compared the procedures against each other. The aim of this study is therefore to compare the three surgical techniques (MACI, AMIC, and MCI) with regard to patient-reported outcomes of pain and function to make recommendations for treatment guidelines in clinical practice.

## 2. Materials and Methods

This retrospective match-pair study included 48 patients who had been treated with AMIC, MACI, or MCI for cartilage lesions in the knee between 2016 and 2021. Local ethical committee of Hamburg approval (2021-10018-BOff) was obtained prior to initiation, and informed consent was provided by all participants prior to being included in the analysis.

### 2.1. Surgical Procedures

#### 2.1.1. MACI

The surgical procedure using MACI involved two main stages:

First, chondral biopsies were harvested arthroscopically from a non-weight-bearing area of the knee using a trephine. Three cylinders were collected and sent to a manufacturing facility, where autologous chondrocytes were expanded in vitro (2–8 million cells/mL). The production process took approximately 24 ± 5 days.

In the second stage, the defect was prepared to ensure a clean, stable, and dry surface. Then, a special hydrogel was applied via a dual-chamber syringe system, simultaneously injecting two components—a chondrocyte suspension and a crosslinker. The hydrogel solidifies in situ within 1–3 min, anchoring the cells without additional fixation. Lastly, the surgeon evaluated implant stability by moving the joint within its physiological range before wound closure. This minimally invasive approach can be performed arthroscopically or via a mini-arthrotomy and ensures precise defect filling with minimal invasiveness.

#### 2.1.2. AMIC

The AMIC procedure was carried out as a one-step procedure. It entailed the insertion of a scaffold into the chondral defect, serving the dual purpose of mechanically stabilizing the cellular clot and encouraging chondrogenic differentiation [13]. An injectable chitosan-based scaffold was used. Chitosan, derived from the deacetylation of chitin found in the exoskeletons of shellfish and insects, offers biocompatibility and biodegradability, making it a suitable scaffold for the proliferation and differentiation of chondrocytes [25,26,27]. In practical terms, the operative technique commences with diagnostic arthroscopy of the knee joint. If necessary, additional procedures, such as meniscal repair and ligament reconstruction, may be performed within the same setting, preceding the cartilage repair.

The preparation of the gel adheres to the techniques outlined in the manufacturer’s product guide. This involves mixing the buffer solution with the chitosan solution and allowing the mixture to stand for 10 min. Subsequently, 5 mL of autologous blood is injected into the prepared buffer–chitosan mixture. This gel blood–buffer–chitosan amalgamation is drawn into an application syringe, ready to be applied to the cartilage lesion.

The procedure continues with the debridement of the cartilage defect site using shavers and curettes. Loose chondral tissue is excised to establish stable vertical walls surrounding the defect. The subchondral plate should be preserved while preparing the defect bed. Microfracture is then performed, involving the use of arthroscopic microfracture awls to create lesions at a depth of approximately 4 mm. These lesions are spaced 3 to 4 mm apart across the surface of the cartilage defect.

Following the drainage of joint fluid, a dry arthroscopy is conducted. The knee’s surface and the cartilage lesion are dried using suction and patties. The injectable scaffold is administered as a viscous gel, targeting horizontal, vertical, and inverted lesions, such as the trochlear, lateral tibial plateau, medial femoral condyle, lateral femoral condyle, and patella. The scaffold is left to solidify within the defect for a period of 5 to 10 min. Subsequently, the knee is mobilized, and wet arthroscopy is repeated to confirm the scaffold’s stability in a fluid-filled environment.

#### 2.1.3. Minced Cartilage

The Minced Cartilage procedure was also performed arthroscopically. The technique was described previously [16]. Symptomatic full-thickness cartilage defects within the knee joint were addressed, such as the trochlea, femoral condyles, tibial plateau, and patella. A clinical examination of the affected knee joint, accompanied by an examination under anesthesia, was considered obligatory.

The patient is usually positioned in a supine posture. The use of a tourniquet is advised to establish a bloodless, dry setting for cartilage implantation. Given the necessity for autologous platelet-rich plasma (PRP) in this procedure, venous blood is drawn from the patient, usually from the cubital veins, under meticulously sterile conditions before anesthesia initiation. This precaution is taken to avert potential adverse effects of narcotic substances on the PRP. Typically, a minimum of 10 to 15 mL of pure PRP is collected, and further processing of PRP takes place during the arthroscopy.

Each indicated Minced Cartilage procedure is initiated via standard arthroscopy of the index knee joint, possibly involving concomitant interventions. The cartilage defect targeted for treatment is thoroughly examined during arthroscopy. The defect is debrided using a small sharp spoon or ringed curette. An optimal debridement technique for cartilage lesions involves the creation of a stable wall and viable rim. Removal of the calcified layer is performed.

The cartilage intended for the procedure can be harvested from the defective cartilage itself, particularly in the case of acute traumatic chondral lesions where the cartilage appears healthy and has recently delaminated. No degenerative cartilage is used for transplantation. The typical and recommended harvest site is the edge of the cartilage defect. If an insufficient amount of cartilage can be collected from the defect’s surroundings, healthy cartilage can be obtained from typical MACI harvest locations, such as the condylar notch. Notably, there is a superior redifferentiation of chondrocytes harvested from the defect’s edge compared to non-weight-bearing regions.

Cartilage is harvested using a 3.0 shaver device connected to a collecting device. Thus, cartilage is harvested, minced into small fragments resembling a paste, and collected immediately. Subsequently, the minced cartilage is mixed with 2 to 3 drops of PRP, resulting in a malleable substance.

The applicator device is loaded with the chips–PRP mixture. Then, 4 mL of PRP is inserted into a specific device to produce autologous thrombin. The chips–PRP paste is applied to the defect to ensure comprehensive coverage. Approximately 50–80% filling, including the surrounding cartilage edge, is typically regarded as sufficient. The consistency of the chip paste provides initial stability. In the subsequent step, thrombin collected from the device is applied drop by drop over the chip paste. The thrombin interacts with the PRP within the chips, generating fibrin that coagulates swiftly, ultimately securing the chips within the cartilage defect. The tissue is sealed with a final layer of fibrin, which is previously mixed, concluding the procedure.

### 2.2. Rehabilitation

Immediately postoperatively, all patients were remobilized in a straight brace for 48 h. On the first day after surgery, a continuous passive motion machine was initiated. For the initial six weeks, patients were allowed partial weight-bearing with crutches, and their range of motion was free. After this six-week period, a gradual progression in weight-bearing and range of motion was authorized, ultimately enabling full weight-bearing and unrestricted range of motion around nine weeks after surgery.

### 2.3. Patient-Related Outcome Measurements

Patient-reported outcome measures (PROMs) were collected before surgery and 6, 12, and 24 months after surgery.

As the primary outcome, the Visual Analogue Scale (VAS) [28], the pain subscale (KOOS-Pain), and the symptoms subscale (KOOS-Symptoms) of the Knee Injury and Osteoarthritis Outcome Score (KOOS) [29,30] were analyzed. These measures directly reflect the knee joint condition. VAS scores range from 0 (indicating no pain) to 10 (indicating the worst pain), and the KOOS-Pain score ranges from 0% (worst pain) to 100% (no pain) and is reflected by 9 out of 42 questions. The KOOS-Symptoms scale is scored the same way, with seven questions counted in the score.

As the secondary outcomes, the KOOS-Activities of daily living (KOOS-ADL) and KOOS-Quality of life (KOOS-QOL), as well as the Tegner Activity Scale (TAS) [31] for assessing the current activity level, were analyzed. These measures reflect the function and activity related to the health status of the joint. The KOOS-Sports and Recreation Subscale was not included because of too many invalid scores over all the time points. The KOOS-ADL and KOOS-QOL are also scored from 0 to 100%, with the ADL asking about 17 usual activities and the QOL including 4 questions with regard to whether the knee joint affects or does not affect the general quality of life.

The TAS is an instrument in which the patient is asked to rate the current level of activity (work, everyday life, and sport) on a scale from 0 to 10. Each of these scales contains a detailed description of the activities, where 0 reflects “Sick leave or disability pension because of knee problems” and 10 “Competitive sports: soccer, football, rugby (national or international elite)”.

Additionally, any postoperative complications and reoperations were documented.

### 2.4. Data Collection and Matching

Data were collected using a prospectively maintained database (Surgical Outcome Measurement System—SOS, Arthrex GmbH, Munich, Germany) that collected patient-related outcome data (PROMs) between 2015 and 2023. The survey was completed electronically, and patients were able to answer the required PROMs online after being sent an E-mail with their individual link to the survey at each time point. Additionally, all patients were reminded by telephone to complete the questionnaires on time.

The data of all patients who underwent one of the three treatments were exported from the system. Patients were eligible if they underwent primary cartilage treatment with no accompanying treatments and had previously agreed to participate in the registry. Other inclusion criteria were not applied.

The patients were matched for gender, age, and BMI, as well as the size and location of the cartilage lesion. In the first step, the age, gender, and BMI of treated patients were filtered to find matching pairs, and second lesion characteristics were distributed as equally as possible between the groups. The matching process started with the smallest group (MACI, *n* = 16) to which one individual of both remaining groups (AMIC: *n* = 19; MCI: *n* = 84) was matched according to the above-mentioned criteria by a person who was blinded to the outcomes.

### 2.5. Statistics

Statistical analysis was performed using IBM SPSS Statistics, Version 27 (IBM Corp., Armonk, NY, USA) and Microsoft Excel, Version 16 (Microsoft Corporation, Redmond, WA, USA). The normal distribution of the data was tested using the Kolmogorov–Smirnov test. ANOVA with repeated measures and Bonferroni-corrected post hoc tests were used to compare VAS and KOOS scores between groups and relevant time points, as well as possible interactions (group*time). Because the requirements for parametric tests were not fulfilled, the Friedmann test was used to compare the TAS scores for each group between all four time points. A single-group repeated measure ANOVA was additionally performed to evaluate within-group differences separately.

In cartilage repair, the patient acceptable state (PASS) is 72.2 for the KOOS-Pain, 71.5 for the KOOS-Symptoms, 86.8 for the KOOS-ADL, and 50.0 for the KOOS-QOL. Further, a difference greater than 8.3 for the KOOS-Pain, 10.7 for the KOOS-Symptoms, 8.8 for the KOOS-ADL, and 18.8 for the KOOS-QOL is considered a clinically important difference (CID) [32]. The minimal clinically important difference (MCID) for the VAS is considered 2.7 [33].

#### Power Estimation

G*Power (Version 3.1.9.7) for ANOVA with repeated measures and within–between interaction was used for power estimation. With the α-level set to 0.05, three groups, four relevant measurements (pre, 6-month, 1-year, 2-year), an estimated correlation among repeated measures of 0.5, and an estimated effect size (f) of 0.2 (η^2^ expected to be 0.04), the statistical power for our planned sample size (48) was estimated to be 0.83 for detecting a medium effect of η^2^ = 0.04.

## 3. Results

Out of the 48 retrospectively included patients, data on all four time points was available for *n* = 12 (MACI), *n* = 13 (AMIC), and *n* =15 (MCI) persons per group. Thus, 40 of 48 datasets were included in the statistical analysis.

Descriptive statistics include the available patient data at each time point. A comprehensive summary of patient attributes and accompanying procedures is presented in Table 1, while details regarding the characteristics of the cartilage defects can be found in Table 2. All included patients underwent isolated cartilage repair surgery with no accompanying intervention.

### 3.1. Prior Treatment

Overall, 13 of the 48 patients (27.1%) had undergone surgery on the index knee within five years prior to the cartilage repair surgery. In the MACI group, three osteochondral surgeries and one tibial realignment osteotomy had been performed. In the AMIC group, one osteochondral treatment and one anterior cruciate ligament reconstruction had previously been performed. In the MCI group, two previous osteochondral treatments, one meniscal refixation, and three ligament reconstructions had been performed (2× ACL/1× MFPL).

### 3.2. Adverse Events

In the MACI group, two people had an indication for a second operation within two years due to ongoing knee pain. In one case, a second surgery was performed 15 months after the treatment due to partly insufficient cartilage coverage in the MFC area. In the other case, the patient complained of ongoing pain one year after the operation, which led to an alternative, non-biological treatment of the cartilage lesion being recommended. This person was lost to follow-up after one year.

In the AMIC group, three patients reported ongoing knee pain. One patient was re-operated after two years because another cartilage lesion had occurred in the trochlea area. Three patients suffered from prolonged knee swelling shortly after surgery. In all cases, this was not caused by any infection and resolved on its own.

In the MCI group, one person reported persistent pain one year after surgery, which was due to another cartilage defect. A second surgery was recommended for this patient. A second person treated with Minced Cartilage in the LFC area reported ongoing pain. The imaging of the area showed sufficient coverage of the lesion and no indication for a second surgery. Lastly, one patient experienced a lateral knee ligament rupture of the index knee one year after surgery.

### 3.3. Patient-Reported Outcome Measurements on Pain and Symptoms

Table 3 shows the descriptive statistics for the VAS, KOOS-Pain, and KOOS-Symptoms.

From pre-surgery to 6 months post-surgery, the VAS scores in all groups decreased. In MCI, the decrement was 2.9 points, in AMIC, it was 1.2 points, and in MACI, it was 1.0 points. From 6 months to 1 year post-surgery, all scores except for the AMIC group (+0.2) further decreased by 0.2 (MCI) and 0.3 (MACI) points. From pre-surgery to the latest follow-up, all three groups decreased their VAS scores by 1.1 (AMIC), 2.4 (MACI), and 2.6 (MCI).

The Pain Subscale of the KOOS increased by 13.5, 5.9, and 19.5 points, respectively, for the AMIC, MACI, and MCI groups from pre- to post-surgery. Overall, two years after surgery, the scores of the KOOS-Pain subscale increased by 13.3 points in the AMIC group, 15.3 points in the MACI group, and 21.6 points in the MCI group compared to pre-surgery.

The KOOS-Symptoms score increased by 8.64, 3.25, and 15.41 points in the AMIC, MACI, and MCI groups, respectively, from pre-surgery to 6 months post-surgery. The maximum score was reached six months post-surgery in the AMIC group (73.5), two years post-surgery in the MACI group (74.6), and two years post-surgery in the MCI group (83.97).

There were no statistically significant group differences in the primary outcomes at baseline (VAS: *p* = 0.075; KOOS-Pain: *p* = 0.73; KOOS-Symptoms: *p* = 0.71).

The ANOVA revealed no significant difference between all three groups for the primary outcomes VAS, KOOS-Pain, and KOOS-Symptoms and no interaction between time and group. However, a significant effect for the factor time was found in both pain-related PROMs: VAS (*p* < 0.001; ES: η^2^ = 0.27) and KOOS-Pain (*p* < 0.001; ES: η^2^ = 0.30).

The post hoc tests showed that for the VAS, the differences between the first time point (pre-surgery) and the other three time points were significant at the 0.05 level (*p* < 0.001; *p* > 0.001; *p* = 0.003). The same was evident for the KOOS-Pain score (*p* = 0.003; *p* < 0.001; *p* < 0.001).

For the VAS, the single-group repeated measure ANOVA revealed that the factor time had a significant effect in the MACI group (*p* = 0.03; ES: η^2^ = 0.24) and the MCI group (*p* < 0.001; ES: η^2^ = 0.46) but not in the AMIC group (*p* = 0.29). For MACI, the difference between pre and 1-year and pre and 2-year were significant (*p* < 0.05). For MCI, all differences between pre and the other time points reached significance (*p* = 0.003; *p* < 0.001; *p* = 0.003).

For the KOOS-Pain, the single-group repeated measure ANOVA revealed that the factor time had a significant effect in all groups (MACI: *p* = 0.01; ES: η^2^ = 0.28; AMIC: *p* = 0.02; ES: η^2^ = 0.23; MCI: *p* < 0.001; ES: η^2^ = 0.40). In MACI, pre to 1-year, and pre to 2-year were significant (*p* = 0.04; *p* = 0.02). In AMIC and MCI, pre was significantly different from all the other time points (AMIC: *p* = 0.04; *p* = 0.02; *p* = 0.04; MCI: *p* = 0.008; *p* = 0.004; *p* < 0.001).

For the KOOS-Symptoms score, the time was also significant (*p* = 0.000; ES: η^2^ = 0.26). Post hoc tests revealed that the differences were significant for the pre-test compared to any other time point (*p* = 0.008; *p* < 0.001; *p* < 0.001).

The single-group repeated measure ANOVA revealed that the factor time had a significant effect in the MACI group (*p* = 0.03; ES: η^2^ = 0.23) and the MIC group (*p* < 0.001; ES: η^2^ = 0.47) but not in the AMIC group (*p* = 0.16). In MACI, only the difference between pre and 2 year was significant (*p* = 0.03) and in MCI the difference between pre and all other time points reached significance (*p* = 0.01; *p* = 0.003; *p* < 0.001).

With the given effect sizes, the statistical test power was calculated to be 1.0 for the factor time in each of the primary outcomes.

Figure 1, Figure 2 and Figure 3 show a comparison of the boxplots for all the groups and time points for the primary outcomes (VAS, KOOS-Pain, KOOS-Symptoms).

### 3.4. Patient-Reported Outcome Measurements on Function and Activity

Descriptive data of the secondary outcomes is displayed in Table 4. The KOOS-ADL score improved by 8.37 points in the AMIC group, 14.34 in the group treated with MACI, and 17.50 in the MCI group from pre-surgery to the 2-year follow-up. Two groups gained the highest improvements from pre-surgery to 6 months post-surgery (AMIC: 4.87; MCI: 8.97). The MACI group gained the highest improvement from 6 months to 1 year post-surgery (10.2 points).

The QOL subscale showed the highest between-group differences at baseline. The AMIC group improved by 13.94 points (from 42.31 to 56.25) from pre-surgery to the 2-year follow-up. The MACI group improved by 22.92 points (from 26.56 to 49.48), and the MCI group by 31.77 (from 29.48 to 61.25).

The TAS was answered by 33 of the 48 included patients at all four time points (AMIC: *n* = 13; MACI: *n* = 8; MIC: *n* = 12). The scores were between 4.25 (MACI) and 4.29 (MCI) pre-surgery and between 3.63 (MACI) and 3.71 (AMIC) at the 2-year follow-up. All scores decreased from pre-surgery to 6 months after surgery (AMIC: −0.87; MACI: −1.92; MCI: −1.1) and approximated the pre-surgery score after two years.

There were no statistically significant group differences detected in the secondary outcomes at baseline (KOOS-ADL: *p* = 0.32; KOOS-QOL: *p* = 0.12; TAS: *p* = 0.80).

For the KOOS-ADL, the ANOVA revealed a highly significant effect of the within-subject factor time point (*p* > 0.000; ES: η^2^ = 0.20). The post hoc tests revealed that the changes between pre-surgery and one year after surgery (*p* = 0.02) and two years after surgery (*p* < 0.000) reached significance.

The single-group repeated measure ANOVA revealed that the factor time had a significant effect in the MACI group (*p* = 0.003; ES: η^2^ = 0.34) and the MCI group (*p* < 0.001; ES: η^2^ = 0.42) but not in the AMIC group (*p* = 0.30). In MACI, only the difference between pre and 1-year and pre and 2-year were significant (*p* = 0.17; *p* = 0.23), and in MCI, the difference between pre and all the other time points reached significance (*p* = 0.005; *p* = 0.002; *p* = 0.002).

The factor time also had a significant effect on the outcomes of the KOOS-QOL (*p* > 0.000; ES: η^2^ = 0.30), with all three follow-up time points being significantly different from pre-surgery scores, respectively (*p* = 0.001; *p* > 0.001; *p* < 0.001).

The single-group repeated measure ANOVA revealed that the factor time had a significant effect in the MACI group (*p* < 0.001; ES: η^2^ = 0.43), and the MCI group (*p* < 0.001; ES: η^2^ = 0.55) but not in the AMIC group (*p* = 0.34). In MACI and MCI, pre was significantly different from all the other time points (AMIC: *p* = 0.002; *p* = 0.009; *p* = 0.003; MCI: *p* = 0.003; *p* < 0.001; *p* < 0.001).

The factor group had no significant effect on either outcome (KOOS-ADL and KOOS-QOL).

With the given effect sizes, the statistical test power was calculated to be 1.0 for the factor time in the KOOS-ADL and KOOS-QOL.

The Friedmann tests did not reveal significant differences between the time points for each group. When conducted as one test for the total sample (*N* = 33), differences in the time points reached statistical significance (*p* < 0.05).

## 4. Discussion

This study aimed to retrospectively compare three procedures that are used for the treatment of cartilage defects in the knee joint. In chronological order, Matrix-Induced Autologous Chondrocyte Implantation (MACI), Autologous Matrix-Induced Chondrogenesis (AMIC), and the Minced Cartilage Technique (MCI), as the most recently developed procedure, were compared in terms of their patient-reported outcomes up to two years after treatment.

All three procedures have shown comparable outcomes in previous studies. However, a direct comparison of all three methods has not yet been carried out. For this reason, a matched-pair analysis was conducted to allow for comparability. The patients were matched for gender, age, BMI, and, when possible, the location and size of the lesion. Other clinical factors, such as prior surgery, lower limb alignment, or ligament stability were not considered in the matching process, although these can have an influence on the outcomes.

Regarding cartilage lesion, location, degree, and size, the MCI group had more grade IV lesions compared to the other groups and the largest mean lesion size. The patients in the MCI group also had the most prior surgeries in sum.

As the most important finding of this study, most of the scores improved within the two years independently of treatment. However, it should be noted that the absence of group differences may also be due to the low statistical test power (.76) for the remaining sample of 40 datasets, which should be further evaluated. The single-group analysis indicated that the AMIC group may be inferior to the others in terms of treatment success, while the MCI group showed the largest effect sizes.

Covering the cartilage defects resulted in a reduction in pain and symptoms for all three treatment groups, as indicated by the descriptive statistics. The patient-reported pain outcomes significantly improved within the first six months to one year after surgery, at least in one of two outcomes. In all the groups, the pain level remained relatively constant thereafter. This is consistent with existing publications. Kaiser et al. [34] showed durable results for the AMIC procedure for up to over 9 years, and Gille et al. [35] showed durable results for the MACI procedure for up to 15 years. Notably, in the single-group analysis, for the AMIC group, significant effects of time were only found in the KOOS-Pain, whereas in the remaining groups (MACI and MCI), time effects were found in all the primary outcomes. This discrepancy may have been caused by the small sample size of only 12 in the AMIC group.

There are currently no comparable results available for the Minced Cartilage Implantation procedure with the arthroscopic technique. For the open procedure, Massen [36] and colleagues also found a significant reduction in pain, as indicated by the Numeric Analog Scale (NAS), after two years. Runer et al. [24] published 5-year results for the open procedure and found a reduction in pain, as indicated by an improvement in the VAS score by 5 points after five years compared to before surgery. The reported results on pain in both studies are consistent with our findings.

In this study, the VAS score was improved by 2.6 points after two years, which is close to the minimal clinically important difference MCID of 2.7 [33] Notably, MCID was reached at one year post-surgery. The VAS score also improved in the other two groups, but their improvements were smaller than in the MCI group. One reason for this may be that the MCI group started from the worst baseline (pre) VAS score and thus had more potential to improve. Although these baseline differences were not significant, this should be investigated further.

For the KOOS-Pain, two groups (AMIC and MIC) exceeded the PASS value of 71.5 [32] after six months. For the MACI group, this was the case after one year. This corresponds to the time point when the clinically important difference (CID) of 8.3 points from baseline was reached in all three groups. Comparing the development of the KOOS-Pain scores, the AMIC group did not further improve after six months, the MACI group improved constantly between all the time points, and the MIC group improved strongly at the beginning and continued to improve afterward. Like the VAS score, the MCI treatment revealed the strongest improvement from pre-surgery to two years after surgery but also started with the worst baseline.

Regarding pain, the VAS and KOOS-Pain show similar results in terms of the magnitude and progress of improvement. Both are pain measures, but the KOOS-Pain is more specific to the retrospectively assessed situations, whereas the VAS is a more unspecific assessment of pain at the time of rating. Therefore, both should be highly correlated.

The symptoms subscale of the KOOS is closely related to the pain subscale, as it also reflects the patients’ evaluations of the joints’ conditions. Therefore, it can be expected that both scales would be highly correlated in terms of their development. In fact, our results revealed that the symptom scores had a similar development compared to the pain scores. The AMIC group nearly reached a plateau after the 6-month time point, the MACI group revealed a constant improvement, and, lastly, the MCI group had the largest improvement from pre-surgery to 6 months post-surgery and further improvements thereafter. Like the results for pain, the MCI group gained the largest increase in the symptoms score compared to the other two groups from pre-surgery to two years after surgery. The PASS of 71.5 [32] was reached after six months in the AMIC and MCI groups and after one year in the MACI group. The CID (10.7) was exceeded by far in the MCI group from pre-surgery to two years post-surgery. In the other two groups, it was slightly missed. This may also be explained by the difference in the baseline values.

Considering the improvement in pain and symptoms, it is not surprising that joint function in everyday situations, as represented by the KOOS-ADL score, also significantly improved after surgery. The KOOS-ADL also asks specific questions about the extent to which a person feels restricted by their affected joint when performing daily activities. Here, the improvement was highest for the MCI group again, followed by MACI and then AMIC. The CID of 8.8 was exceeded in the MACI (1 year) and MCI (6 months) groups but slightly missed (<0.5 points) in the AMIC group. The PASS of 86.8 was reached only in the MCI group and nearly reached in the other two groups. Again, it is important to consider the differences in the baseline scores when comparing the improvements between the three groups. Overall, each of the treatments resulted in significant improvements in daily functioning of the knee joint. A tendency towards faster recovery of function can be noted after MCI treatment.

Pain and limited joint function severely affect the overall quality of life (QOL) of an individual. Cartilage repair, like almost every medical treatment, aims to improve the quality of life for the patient. The KOOS-QOL, therefore, was also assessed. All three groups were severely affected in QOL by their joints before surgery, as indicated by the sores below 50 points. The patients in all three groups exceeded the PASS of 50.0 points or nearly reached it. The CID of 18.8 was reached by the MACI group (2 years) and the MCI group (1 year) but was slightly missed by the AMIC group. The AMIC group, however, started from a baseline of 42.31, far higher than the other two groups (MACI: 26.56; MCI: 29.48). Since KOOS sub-scores are highly correlated [29], the magnitude of the baseline difference for the QOL subscale was as surprising as the fact that this time, the AMIC group was the outlier.

Lastly, we aimed to evaluate how activity levels changed throughout the different treatments. Here, we evaluated the TAS-Score, which presents the patients with different activity levels to choose from. As evident from the statistical analysis, we could not show significant differences between all four time points. Furthermore, the score more likely decreased from baseline. This measure represents a ranking from 0 to 10, with each number representing a certain state of activity. Considering this fact, scores must be seen as activity stages. The activity state, therefore, recovered to the baseline level (stage 4) after one year in AMIC and MCI and after two years in MACI. Comparably, Gille et al. [35] found that the Tegner score improved from 3.0 to 3.6 after five years and further to 5.2 after five years, which indicates that the activity level may further improve in our participants.

Overall, our results are comparable with previous studies in terms of improvements in pain and function after surgery.

### Limitations

This study was the first to directly compare three different competing procedures to make treatment recommendations for clinical practice. It must, however, be admitted that there is some limitation in the predictive value caused by the difference in baseline values (although not significant) and the retrospective design of this study. The test power with 40 participants remaining for the analysis may have been too small to determine possible group differences. Furthermore, female participants are underrepresented in this study.

Our matching process was performed manually instead of using a more rigorous mathematical approach like propensity analysis. Despite these intensive efforts in the matching process, the location and size of the lesion were not 100% equally distributed. This study also did not control for other clinical factors (e.g., previous treatment, lower limb alignment, ligament stability) that can have an influence on the outcomes. This addition would allow a more precise assessment of the effectiveness of therapy.

The last limitation that must be discussed is the lack of outcome measures other than the patient-reported outcomes. The addition of more objective data, such as clinical mobility or strength measurements, would contribute to a better understanding of the functional differences between the three methods. The objective assessment of cartilage regeneration via magnetic resonance imaging (MRI) or vibratrography, which allows for dynamic assessment of joint function, should supplement future work.

Overall, a prospective approach with a larger sample size that includes female and male participants equally, considers the baseline values of the main outcomes, as well as all accompanying clinical factors, and utilizes subjective and objective outcomes would be highly recommended for future studies.

## 5. Conclusions

This study shows that all three procedures (AMIC, MACI, MCI) are recommendable for the treatment of cartilage lesions in the knee joint. The appropriate procedure can therefore be chosen according to the resources, skills, and preferences of the surgeon and the clinic. MCI may provide an advantage in recovery time and overall treatment success. All three procedures will likely lead to improvements in pain, function, and quality of life for the patient.

## Figures and Tables

**Figure 1 jcm-14-02194-f001:**
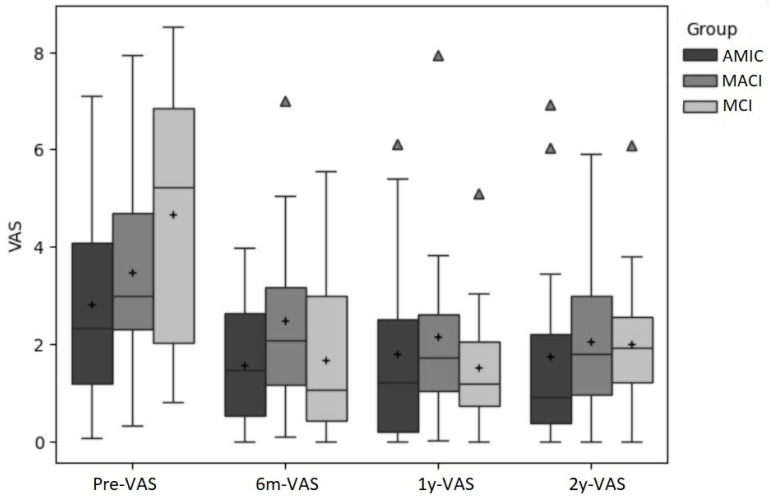
VAS scores for all three groups for all four time points. “+” indicates the mean, and outliers are indicated by triangles.

**Figure 2 jcm-14-02194-f002:**
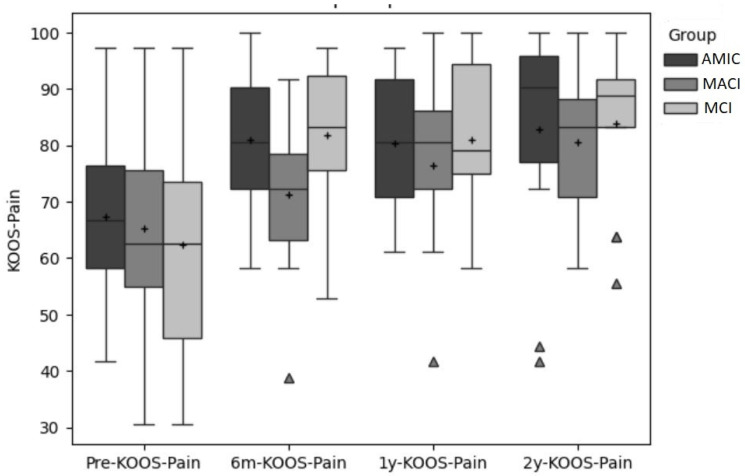
KOOS-Pain scores for all three groups for all four time points. “+” indicates the mean, and outliers are indicated by triangles.

**Figure 3 jcm-14-02194-f003:**
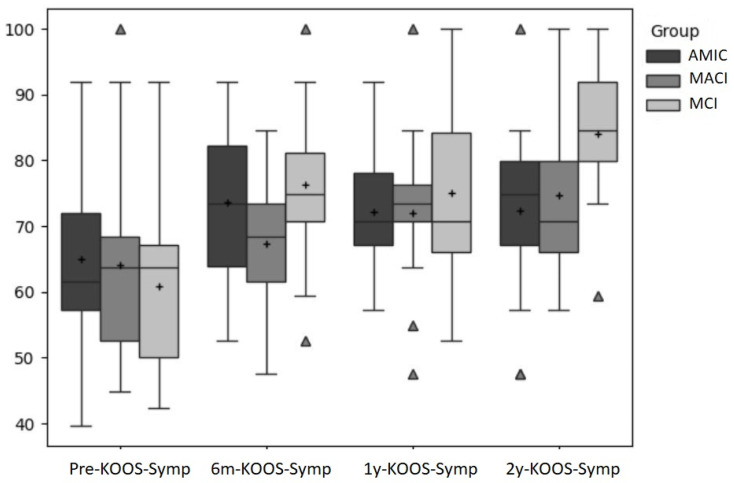
KOOS-Symptoms scores for all three groups for all four time points. “+” indicates the mean, and outliers are indicated by triangles.

**Table 1 jcm-14-02194-t001:** Patient characteristics mean ± SD of *N* = 48 persons.

	MACI	AMIC	MCI
gender	1 f/15 m	1 f/15 m	1 f/15 m
age at treatment	36.9 ± 7.9	36.2 ± 9.2	36.7 ± 9.1
height (cm)	184.4 ± 7.6	180.6 ± 8.4	180.9 ± 7.0
weight (kg)	90.5 ± 9.9	91.2 ± 21.8	89.4 ± 15.3
BMI	26.7 ± 3.3	27.8 ± 5.5	27.3 ± 4.5

**Table 2 jcm-14-02194-t002:** Location and size of cartilage lesions of *N* = 48 persons.

Location	*n* MACI	*n* AMIC	*n* MCI
medial FC	6	7	7
lateral FC	3	4	3
med + lat FC	2	1	2
trochlear	3	2	3
retropatellar	2	1	1
retropatellar + MFC		1	
Grade			
IV	9	9	11
III–IV	4	5	3
III	3	1	2
Size (cm^2^)			
<1–1	2	2	3
>1–2	5	8	2
>2–3	5	3	7
>3	4	4	4
Mean	2.5 ± 1.2	2.3 ± 0.9	2.6 ± 1.1

**Table 3 jcm-14-02194-t003:** Descriptive data for the primary outcomes (VAS; KOOS-Pain; KOOS-Symptoms) for all four time points (mean ± SD).

VAS	pre	6 Months	12 Months	24 Months
MACI	3.5 ± 2.1	2.5 ± 1.9	2.2 ± 2.0 *	2.1 ± 1.7 *
AMIC	2.8 ± 2.1	1.6 ± 1.3	1.8 ± 1.9	1.7 ± 2.2
MCI	4.6 ± 2.5	1.7 ± 1.7 *	1.5 ± 1.3 **	2.0 ± 1.6 *
KOOS-Pain				
MACI	65.28 ± 17.82	71.18 ± 13.18	76.50 ± 14.81 *	80.56 ± 13.08 *
AMIC	67.41 ± 15.51	80.93 ± 12.89 *	80.37 ± 11.57 *	80.74 ± 18.85 *
MCI	62.34 ± 19.51	81.77 ± 13.91 *	80.90 ± 12.98 *	83.89 ± 12.91 **
KOOS-Symptoms				
MACI	63.97 ± 15.37	67.22 ± 9.44	71.96 ± 12.76 *	74.60 ± 14.30 *
AMIC	64.86 ± 13.10	73.50 ± 11.95	72.13 ± 10.42	72.39 ± 14.34
MCI	60.81 ± 13.76	76.22 ± 12.44 *	74.95 ± 14.43 *	83.97 ± 11.05 **

* significant effect of time *p* ≤ 0.005; ** high significant effect of time *p* ≤ 0.001.

**Table 4 jcm-14-02194-t004:** Descriptive data for the primary outcomes (VAS; KOOS-Pain; KOOS-Symptoms) for all four time points (mean ± SD).

KOOS-ADL	Pre	6 Months	12 Months	24 Months
MACI	72.30 ± 15.99	77.33 ± 16.45	87.50 ± 10.48 *	86.64 ± 11.79 *
AMIC	77.94 ± 16.60	82.81 ± 13.60	80.32 ± 15.71	86.31 ± 19.29
MCI	70.25 ± 18.64	79.22 ± 13.88	86.37 ± 12.06 *	87.75 ± 12.96 *
KOOS-QOL				
MACI	26.56 ± 16.88	40.62 ± 15.45 *	44.27 ± 19.67 *	49.48 ± 16.74 *
AMIC	42.31 ± 23.41	52.40 ± 20.97	54.32 ± 23.99	56.25 ± 31.15
MCI	29.48 ± 11.68	47.08 ± 20.45 *	53.75 ± 19.16 **	61.25 ± 18.63 **
TAS				
MACI	4.25 ± 2.60	2.33 ± 1.64	3.00 ± 1.12	3.63 ± 1.51
AMIC	4.27 ± 2.12	3.40 ± 1.64	3.87 ± 1.88	3.71 ± 1.64
MCI	4.29 ± 2.92	3.19 ± 1.38	4.00 ± 1.60	3.69 ± 2.02

* significant effect of time *p* ≤ 0.005; ** high significant effect of time *p* ≤ 0.001.

## Data Availability

The raw data supporting the conclusions of this article will be made available by the authors on request.

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
