# Peer review of "Comparison of Three Different Techniques for the Treatment of Cartilage Lesions—Matrix-Induced Autologous Chondrocyte Implantation (MACI) Versus Autologous Matrix-Induced Chondrogenesis (AMIC) and Arthroscopic Minced Cartilage—A 2-Year Follow-Up on Patient-Reported Pain and Functional Outcomes"

_jcm, 2025, doi:10.3390/jcm14072194_

Round 1
Reviewer 1 Report
Comments and Suggestions for Authors
Overall, the article is well-written, and the experiment is designed rigorously. However, several concerns should be addressed:
- In the introduction, such as MFX, a brief introduction (maybe one sentence) can be added to improve clarity.
- Page 2, line 76: Is this method supported by high-quality clinical research or meta-analysis? A reference should be included.
- There are some typos in the manuscript, such as on page 2, line 83. The citation format frequently contains errors—please check and correct them.
- In the methods section, how was the sample size determined? Was a power analysis conducted to ensure sufficient statistical power?
- Were there any inclusion or exclusion criteria in the study?
- For the surgical procedures, were they all performed by the same surgeon(s) to minimize variability?
- Are the baseline differences in outcomes statistically significant? Could they introduce bias in the study? (As mentioned in the discussion, page 13, lines 438-440.)
- Page 6, line 216: How were these time points selected?
- The statistical significance results should also be labeled in the tables.
- References for all the scales used in the paper should be provided.
- Why are some cases missing? Could this lead to any bias?
- In the main text, abbreviations should only appear once.
Author Response
Dear reviewer, thank you for taking the time to review and comment our manuscript. You have provided an extensive and detailed review, which has helped us a lot in improving our manuscript. Based on your comments, we conducted further and informative statistical analysis that we included into the manuscript. We have carefully addressed and revised each single point raised. You can find the answers and corresponding changes listed under each comment.
Overall, the article is well-written, and the experiment is designed rigorously. However, several concerns should be addressed:
1) In the introduction, such as MFX, a brief introduction (maybe one sentence) can be added to improve clarity.
Reply: Thank you for this note. We have added more information in the paragraph: “Microfracture (MFX) has been widely used in patients since 1998 and involves creating perforations in the subchondral bone, spaced 3 to 4 mm apart, to release bone marrow components into the defect. To date, MFX stands as a widely adopted bone marrow stimulation strategy, and a range of cell-based restoration techniques utilizing adult or juvenile, autograft or allograft cartilage sources, along with chondrocyte and nonchondrocyte options, are now available”. (p. 2 ll. 82-87)
2) Page 2, line 76: Is this method supported by high-quality clinical research or meta-analysis? A reference should be included.
Reply: We apologize for not having provided sufficient references for MACI. We have added a late systematic literature review, evidence grade 4, that comprises the latest research and supports excellent outcomes in the long term for this method:
Wang AS, Nagelli CV, Lamba A, Saris DBF, Krych AJ, Hevesi M. Minimum 10-Year Outcomes of Matrix-Induced Autologous Chondrocyte Implantation in the Knee: A Systematic Review. The American Journal of Sports Medicine. 2024;52(9):2407-2414. doi:10.1177/03635465231205309
Further we have added another source to support the positive evaluation in 2005 and assigned the original source 5 to the right year (2017).
We rephrased the section for a better chronological order and updated the sources:
“Historically, the gold standard for repairing cartilage defects has been Matrix-induced Autologous Chondrocyte Implantation (MACI). Dr. Lars Peterson pioneered this treatment method in 1985, which initially faced skepticism but later achieved notable successes in 1989. In MACI, cartilage cells are harvested in a first surgical procedure, cul-tivated in the laboratory, and then implanted in a second procedure. The two-step MACI approach incurs significant costs associated with multiple surgeries and cell expansion in the laboratory. In 2000, the United Kingdom's National Institute for Health and Care Ex-cellence did not recommend routine use of the procedure due to a lack of well-designed studies. In 2005 and later, in 2017, positive clinical and economical evaluations were available. MACI's ability to improve clinical outcomes is now supported by numer-ous studies including a minimum 10-years outcome systematic review conducted in 2024.” (p.2-3, ll. 88-97)
3) There are some typos in the manuscript, such as on page 2, line 83. The citation format frequently contains errors—please check and correct them.
Reply: Thank you for spotting this. We have changed the Year into 2005 (p. 3, l.104). We have also corrected other errors we found.
4) In the methods section, how was the sample size determined? Was a power analysis conducted to ensure sufficient statistical power?
Reply: Since we only had 16 patients in the smallest group, the overall sample size of 48 was determined by this group. Power calculation revealed statistical power of .83 for our given sample size of 48 patients for detecting an small to medium effect of â´„² = .04. This was stated in our manuscript p. 7, ll. 263 – 267. We have now added a subheading to this part (2.5.1 power estimation) (p. 7, l. 292). This calculation was done before the real sample size (participants with data for all time-points) and effect sizes were calculated.
Considering one of the later points of the reviewer, we have additionally calculated the statistical power post-hoc for each singe outcome based on the given sample size of 40 and the respective effect sizes and added this to the results:
“With the given effect sizes, statistical test power was calculated to be 1.0 for the factor time in each of the primary outcomes.” (p. 10, ll. 403-404)”
“With the given effect sizes, statistical test power was calculated to be 1.0 for the factor time in KOOS-ADL and KOOS-QOL.” (p. 12; ll. 453-454)
5) Were there any inclusion or exclusion criteria in the study?
Reply: This is a very good point raised by the reviewer. Since this was a retrospective study, we did not state any inclusion criteria explicitly. One of the main criteria was that the cartilage surgery was performed as a single procedure without additional treatments (p. 7, lines 306-307). Further it had to be a primary treatment and no revision treatment. We did not have any criteria on age as we wanted to comprise the variability. Prior treatments were also not excluded but reported. Other criteria such as limb alignment or joint stability would be recommended for the future.
We have added a statement to make this more comprehensible:
“Patients were found eligible if they underwent primary cartilage treatment with no companying treatments and former agreed on participation in the registry. Other inclusion criteria were not applied” (p. 6, ll. 267 – 269)
6) For the surgical procedures, were they all performed by the same surgeon(s) to minimize variability?
Reply: This is also a valid point. The retrospective character and participant number of the study would not allow for only a single surgeon. In this case, three surgeons performed the surgeries. The fact that those surgeries were performed in separate time windows due to their availability with the MIC approach being the latest one the main surgeon of this one was another then of the previous two. However, objectivity is guaranteed by strict operating instructions of the clinic and the guideline of each approach followed by the surgeon.
7) Are the baseline differences in outcomes statistically significant? Could they introduce bias in the study? (As mentioned in the discussion, page 13, lines 438-440.)
Reply: Thank you for this question. We have calculated a one-factor ANOVA for each respective outcome to check for baseline differences. There was no statistically significant group difference in all outcomes at baseline (VAS: p = .075; KOOS-Pain: p = .73; KOOS-Symptoms: p = .71; KOOS-ADL: p = .32; KOOS-QOL: p = .12; TAS: p = .80) We have added this information to the results section. We have added the information in the manuscript. (p. 9 ll. 376-377; p. 12, ll. 433-434)
8) Page 6, line 216: How were these time points selected?
Reply: The time points were preselected by the company providing the outcomes registry. We could also have used more prior time points (e.g. 3 month, 6 weeks). However, with every added time- point, more participants would fail the requirement of having data for all time points available.
9) The statistical significance results should also be labelled in the tables.
Reply: This point has brought our attention to the fact that we did not evaluate the groups separately with regard on the within effects at different time points. We have conducted further analysis and added the respective results. (p. 10, ll. 385 ff; p. 12 ll. 439 ff)
We have also added the requested labels in table 3 (p. 9) and 4 (p. 12).
Thank you for spotting this.
10) References for all the scales used in the paper should be provided.
Reply: Thank you for the important note. We have provided references for all scales that were used.
KOOS :
Collins, N J, Prinsen, C A C, Christensen, R, Bartels, E M, Terwee, C B, & Roos, E M. Knee Injury and Osteoarthritis Outcome Score (KOOS): systematic review and meta-analysis of measurement properties. Osteoarthritis and cartilage, 2016, 24. Jg., Nr. 8, S. 1317-1329.
Bekkers, J E J. et al. Validation of the Knee Injury and Osteoarthritis Outcome Score (KOOS) for the treatment of focal cartilage lesions." Osteoarthritis and cartilage 17.11 (2009): 1434-1439.
VAS for knee osteoathrosis:
Alghadir, A. H., Anwer, S., Iqbal, A., & Iqbal, Z. A. (2018). Test–retest reliability, validity, and minimum detectable change of visual analog, numerical rating, and verbal rating scales for measurement of osteoarthritic knee pain. Journal of Pain Research, 11, 851–856.
Tegner German Translation:
Wirth, B., et al. "Development and evaluation of a German version of the Tegner activity scale for measuring outcome after anterior cruciate ligament injury." Sportverletzung Sportschaden: Organ der Gesellschaft fur Orthopadisch-Traumatologische Sportmedizin 27.1 (2013): 21-27.
11) Why are some cases missing? Could this lead to any bias?
Reply: The missing cases are those subjects who did not answer all the four timepoints. In our statistical analysis, we did not replace missing values. Therefore, the number of participants dropped to 40, decreasing the statistical power and therefore the possibility of undetected differences. This is especially relevant for the group differences. It is possible that these were not detected because of the decreased test power of .76 leading to a chance of 24 % of undetected differences. We have added this to the discussion and limitations section.
“However, it should be noted that the absence of group differences may also be due to the low statistical test power (.76) for the remaining sample of 40 data sets and should be further evaluated. The single group analysis indicates that the AMIC group may be inferior to the others in terms of treatment success while the MCI group showed the largest effect sizes.” (p. 13, ll. 481 - 485).
“The test power with the 40 participants remaining for the analysis may have been too small to determine possible group differences.” (p. 15, ll. 576-578)
We prefer this over the replacement of missing values by means of the group or other algorithms. However, we are open to discuss other solutions.
12) In the main text, abbreviations should only appear once.
Reply: Thank you for the advice. We have understood that we use the full term once and thereafter only the abbreviations. We have changed the main text accordingly. Just in the discussion, we prefer to shortly introduce the Abbreviations again to make it easier for the readers.
Reviewer 2 Report
Comments and Suggestions for Authors
The study is interesting, but the matching process is arbitrary. A more rigorous approach, such as propensity analysis, would be preferable. If needed, pairwise comparisons should be made, with the minced group—the largest—separately matched against AMIC and MACI. For AMIC vs. MACI, matching is unnecessary, as multivariate analyses and post-hoc tests are sufficient to highlight differences.
The study is prospective but non-randomized, as data collection was conducted prospectively in the registry.
Author Response
Dear reviewer, thank you for taking the time to review and comment our manuscript. You have provided some valid points that we needed to consider and to address. You can find our answers and changes made to the manuscript based on your comments listed below.
The study is interesting, but the matching process is arbitrary. A more rigorous approach, such as propensity analysis, would be preferable. If needed, pairwise comparisons should be made, with the minced group—the largest—separately matched against AMIC and MACI. For AMIC vs. MACI, matching is unnecessary, as multivariate analyses and post-hoc tests are sufficient to highlight differences.
Reply:
Thank you for the note on our matching process. We agree that propensity analysis would possibly lead to better objectivity and is statistically sounder. In this case, we decided to do it manually because we utilized a hierarchical approach. Matching gender and age were prioritized at the first level and the lesion size and region at second. We have discussed limitations of our approach in the manuscript, and we have added this point. “Our matching process was done manually instead of using a more rigorous mathematical approach like propensity analysis.” (p. 15, ll. 580 – 581)
Additionally, as recommended by reviewer 1, we have added group wise comparisons of baseline-differences to detect possible bias caused by the baseline differences. Baseline differences were not found. (p. 9 ll. 376-377; p. 12, ll. 433-434)
The study is prospective but non-randomized, as data collection was conducted prospectively in the registry.
Reply: We agree with the reviewer that the data were collected prospectively by a registry. This is stated in the manuscript (p. 6, line 260-262). However, the hypothesis was not defined prior to data collection. Furthermore, there was no initial study protocol and no pre-registration or a priori estimation of sample size. The matching was also conducted retrospectively. Therefore, we would not consider this a prospective study (which downgrades our evidence level) as we did not fulfil all these requirements.
Reviewer 3 Report
Comments and Suggestions for Authors
The article addresses the important topic of comparing three methods of treating articular cartilage damage: MACI, AMIC and MIC. Its strengths are the retrospective analysis of matched pairs of patients and the use of reliable scales for assessing pain and joint function (VAS, KOOS, TAS). However, there are some methodological limitations and issues regarding the interpretation of the results that should be refined.
The article is well written and based on sound research methods. However, several aspects need to be improved to increase clarity, quality of interpretation of results and potential application in clinical practice. I provide detailed comments below.
Minor comments:
The introduction is too short, please complete and introduce more detailed information about articular cartilage, injuries and changes in parameters related to degeneration. Please complete using the latest references:
https://doi.org/10.1016/j.jmbbm.2024.106575
DOI 10.3390/healthcare12161648
The paper focuses on patient-reported outcomes, but does not include objective imaging data (e.g., MRI) on cartilage regeneration. I strongly recommend that future studies be supplemented with MRI imaging data this will allow an objective assessment of cartilage regeneration, which would significantly increase the diagnostic value of the article. In addition, alternative methods of cartilage damage diagnosis such as vibratrography that allows dynamic assessment of joint function may appear useful here. Please discuss this method briefly in the discussion and consider its use in further research.
The size of the groups (MACI: 16, AMIC: 16, MIC: 16) and the fact that the final analysis included only 40 patients may affect the statistical power of the study. In my opinion, expanding the study cohort would improve the statistical power of the results and allow for a more robust analysis of the effectiveness of the methods. Please consider this in the limitations and plans for future studies.
The article does not sufficiently consider the potential influences of previous surgeries and other clinical factors (e.g., degree of cartilage degeneration, lower limb alignment, ligament stability). Consideration of additional clinical factors, such as previous surgeries or axial deformities, may influence a more precise assessment of the effectiveness of therapy please bring up this topic in the discussion.
The observation period is 2 years, which in my opinion is a relatively short period for cartilage regenerative methods. It would be worthwhile to extend the observation period to a minimum of 5 years to obtain a more comprehensive evaluation of the long-term effectiveness of the methods. Please consider this in your plans for further studies.
Although the authors mention differences in the costs of the methods, they do not provide a detailed economic analysis of them. Expanding the analysis of treatment costs and their economic effectiveness could facilitate clinical decisions regarding the choice of treatment method. Making appropriate additions will significantly enhance the applied nature of the work.
Author Response
Dear reviewer, thank you for taking the time to review and comment our manuscript. You have provided a well elaborated review, which has helped us a lot. Based on your comments, we had the chance to improve the introduction and discussion and consider some limitations. We have carefully addressed and revised each single point raised. You can find the answers and corresponding changes listed under each comment.
Minor comments:
The introduction is too short, please complete and introduce more detailed information about articular cartilage, injuries and changes in parameters related to degeneration. Please complete using the latest references:
https://doi.org/10.1016/j.jmbbm.2024.106575
DOI 10.3390/healthcare12161648
Reply: Thank you for bringing this to our attention. We agree that we currently have deficits regarding the properties and changes in articular cartilage. However, as this is a brief introduction, we would prefer not to go into too much depth in this section. We have extended this part based on your suggestion and literature. (p. 2, ll. 49-68)
The paper focuses on patient-reported outcomes, but does not include objective imaging data (e.g., MRI) on cartilage regeneration. I strongly recommend that future studies be supplemented with MRI imaging data this will allow an objective assessment of cartilage regeneration, which would significantly increase the diagnostic value of the article. In addition, alternative methods of cartilage damage diagnosis such as vibratrography that allows dynamic assessment of joint function may appear useful here. Please discuss this method briefly in the discussion and consider its use in further research.
Reply: We absolutely agree with the reviewer in this point. We already stated briefly in the last part of the limitations section of the manuscript that the study would benefit from objective MRI data. “The addition of more objective data, such as clinical mobility or strength measurements and imaging data (MRI), would contribute to a better understanding of the differences between the three methods.”
We have extended this section with a clear recommendation for future work:
“The last limitation that must be discussed is the lack of outcome measures other than the patient reported outcomes. The addition of more objective data, such as clinical mobility or strength measurements would contribute to a better understanding of the functional differences between the three methods. Especially the objective assessment of cartilage re-generation via magnetic resonance imaging (MRI) or vibratrography which allows for dynamic assessment of joint function should supplement future work.
Overall, a prospective approach that includes female and male participants equally, considers the baseline values of the main outcomes as well as prior treatment and utilizes subjective and objective outcomes would be highly recommended for future studies.” (p. 15 ll. 586-595).
The size of the groups (MACI: 16, AMIC: 16, MIC: 16) and the fact that the final analysis included only 40 patients may affect the statistical power of the study. In my opinion, expanding the study cohort would improve the statistical power of the results and allow for a more robust analysis of the effectiveness of the methods. Please consider this in the limitations and plans for future studies.
Reply: This is a valid pint raised by the reviewer. We have added the consideration of this study being underpowered in the discussion and limitations. We believe that the study was underpowered for finding group differences. We have added this consideration in the discussion and limitations:
“However, it should be noted that the absence of group differences may also be due to the low statistical test power (.76) for the remaining sample of 40 data sets and should be further evaluated. The single group analysis indicates that the AMIC group may be inferior to the others in terms of treatment success while the MCI group showed the largest effect sizes.” (p. 13, ll. 481 - 485).
“The test power with the 40 participants remaining for the analysis may have been too small to determine possible group differences.” (p. 15, ll. 576-578)
The article does not sufficiently consider the potential influences of previous surgeries and other clinical factors (e.g., degree of cartilage degeneration, lower limb alignment, ligament stability). Consideration of additional clinical factors, such as previous surgeries or axial deformities, may influence a more precise assessment of the effectiveness of therapy please bring up this topic in the discussion.
Reply: Thank you for the note. The reviewer is right in that we did not sufficiently consider accompanying clinical factors in this study. The degree of cartilage degeneration was part of the sample matching and description. Prior surgeries were retrospectively reported but not considered in advance or in the discussion. We extended the discussion and limitations according to the advice.
“The patients were matched for gender, age, BMI and, when possible, the location and size of the lesion. Other clinical factors, such as prior surgery, lower limb alignment or ligament stability were not considered in the matching process although these can have an influence on the outcomes.” Regarding cartilage lesion, location, degree and size, the MCI group has more grade IV lesions compared to the other groups and the largest mean lesion size. Patients of the MCI group also had the most prior surgeries in sum.” (p. 13, ll. 473 – 479)
“This study also lacked controlling other clinical factors (e.g. previous treatment, lower limb alignment, ligament stability) which can have an influence on the outcomes. This addition would allow a more precise assessment of the effectiveness of therapy” (p. 15, ll. 582 – 585)
The observation period is 2 years, which in my opinion is a relatively short period for cartilage regenerative methods. It would be worthwhile to extend the observation period to a minimum of 5 years to obtain a more comprehensive evaluation of the long-term effectiveness of the methods. Please consider this in your plans for further studies.
Reply: Yes, we agree that an extension of the observation period to at least five years would be desirable. We are currently extending data collection as we are aiming for a 5 year follow up.
Although the authors mention differences in the costs of the methods, they do not provide a detailed economic analysis of them. Expanding the analysis of treatment costs and their economic effectiveness could facilitate clinical decisions regarding the choice of treatment method. Making appropriate additions will significantly enhance the applied nature of the work.
Reply: We understand the point raised by the reviewer. We have shortly mentioned the economic perspective but not elaborated this point. Although cost evaluation this is important in clinical practice, we believe that an economic perspective goes beyond the main topic of our study. Here, we focussed on the outcomes for the patient. We have therefore removed the mention of costs and left it to other experts in this field to assess it more precisely.
Removed sentences:
“The two-step MACI approach incurs significant costs associated with multiple surgeries and cell expansion in the laboratory.” (p. 2)
“Nonetheless, AMIC offers several notable advantages, being more effective than MFX and more cost-effective than MACI [13].” (p. 3)